# Semi-Supervised Learning of Multi-Object 3D Scene Representations

## Abstract

Representing scenes at the granularity of objects is a prerequisite for scene under-standing and decision making. We propose a novel approach for learning multi-object 3D scene representations from images. A recurrent encoder regresses a latent representation of 3D shapes, poses and texture of each object from an input RGB image. The 3D shapes are represented continuously in function-space as signed distance functions (SDF) which we efficiently pre-train from example shapes in a supervised way. By differentiable rendering we then train our model to decompose scenes self-supervised from RGB-D images. Our approach learns to decompose images into the constituent objects of the scene and to infer their shape, pose and texture from a single view. We evaluate the accuracy of our model in inferring the 3D scene layout and demonstrate its generative capabilities.

## 1 Introduction

Humans have the remarkable capability to decompose scenes into its constituent objects and to infer object properties such as 3D shape and texture from just a single view. Providing intelligent systems with similar capabilities is a long-standing goal in artificial intelligence. Such representations would facilitate object-level description, abstract reasoning and high-level decision making. Moreover, object-level scene representations could improve generalization for learning in downstream tasks such as robust object recognition or action planning.

Previous work on learning-based scene representations focused on single-object scenes (Sitzmann et al., 2019) or neglected to model the 3D geometry of the scene and the objects explicitly (Burgess et al., 2019; Greff et al., 2019; Eslami et al., 2016). In our work, we propose a multi-object scene representation network which learns to decompose scenes into objects and represents the 3D shape and texture of the objects explicitly. Shape, pose and texture are embedded in a latent representation which our model decodes into textured 3D geometry using differentiable rendering. This allows for training our scene representation network in a semi-supervised way. Our approach jointly learns the tasks of object detection, instance segmentation, object pose estimation and inference of 3D shape and texture in single RGB images. Inspired by (Park et al., 2019; Oechsle et al., 2019; Sitzmann et al., 2019), we represent 3D object shape and texture continuously in function-space as signed distance and color values at continuous 3D locations. The scene representation network infers the object poses and its shape and texture encodings from the input RGB image. We propose a novel differentiable renderer which efficiently generates color and depth images as well as instance masks from the object-wise scene representation. By this, our model facilitates to generate new scenes by altering an interpretable latent representation (see Fig. 1). Our network is trained in two stages: In a first stage, we train an auto-decoder subnetwork of our full pipeline to embed a collection of meshes in continuous SDF shape embeddings as in DeepSDF (Park et al., 2019). With this pre-trained shape space, we train the remaining parts of our full multi-object network to decompose and describe the scene by multiple objects in a self-supervised way from RGB-D images. No ground truth of object pose, shape, texture, or instance segmentation is required for the training on multi-object scenes. We denote our learning approach semi-supervised due to the supervised pre-training of the shape embedding and the self-supervised learning of the scene decomposition.

We evaluate our approach on synthetic scene datasets with images composed of multiple objects to show its capabilities with shapes such as geometric primitives and vehicles and demonstrate the properties of our geometric and semi-supervised learning approach for scene representation. In sum-

Figure 1: **Example scenes with object manipulation.** For each example, we input the left images and compute the middle one as standard reconstruction. After the manipulation in the latent space, we obtain the respective right image. Plausible new scene configurations are shown on the Clevr dataset (Johnson et al., 2017) (top) and on composed ShapeNet models (Chang et al., 2015) (bottom).

mary, we make the following **contributions**: **(1)** We propose a novel model to learn representations of scenes composed of multiple objects. Our model describes the scene by explicitly encoding object poses, 3D shapes and texture. To the best of our knowledge, our approach is the first to jointly learn the tasks of object instance detection, instance segmentation, object localization, and inference of 3D shape and texture in a single RGB image through self-supervised scene decomposition. **(2)** Our model is trained by using differentiable rendering for decoding the latent representation into images. For this, we propose a novel differentiable renderer using sampling-based raycasting for deep SDF shape embeddings which renders color and depth images as well as instance segmentation masks. **(3)** By representing 3D geometry explicitly, our approach naturally respects occlusions and collisions between objects and facilitates manipulation of the scene within the latent space. We demonstrate properties of our geometric model for scene representation and augmentation, and discuss advantages over multi-object scene representation methods which model geometry implicitly. We plan to make source code and datasets of our approach publicly available upon paper acceptance.

## 2 RELATED WORK

**Deep learning of single object geometry.** Several recent 3D learning approaches represent single object geometry by implicit surfaces of occupancy or signed distance functions which are discretized in 3D voxel grids (Kar et al., 2017; Tulsiani et al., 2017; Wu et al., 2016; Gadelha et al., 2017; Qi et al., 2016; Jimenez Rezende et al., 2016; Choy et al., 2016; Shin et al., 2018; Xie et al., 2019). Voxel grid representations typically waste significant memory and computation resources in scene parts which are far away from the surface. This limits their resolution and capabilities to represent fine details. Other methods represent shapes with point clouds (Qi et al., 2017; Achlioptas et al., 2018), meshes (Groueix et al., 2018), deformations of shape primitives (Henderson & Ferrari, 2019) or multiple views (Tatarchenko et al., 2016). In continuous function-space representations, deep neural networks are trained to directly predict signed distance (Park et al., 2019; Xu et al., 2019; Sitzmann et al., 2019), occupancy (Mescheder et al., 2019; Chen & Zhang, 2019), or texture (Oechsle et al., 2019) at continuous query points. We use such representations for individual objects.

**Deep learning of multi-object scene representations.** Self-supervised learning of multi-object scene representations from images recently gained significant attention in the machine learning community. MONet (Burgess et al., 2019) presents a multi-object network which decomposes the scene using a recurrent attention network and an object-wise autoencoder. It embeds images into object-wise latent representations and overlays them into images with a neural decoder. Yang et al. (2020) improve upon this work. Greff et al. (2019) use iterative variational inference to optimize object-wise latent representations using a recurrent neural network. SPAIR (Crawford & Pineau, 2019) and SPACE (Lin et al., 2020) extend the attend-infer-repeat approach (Eslami et al., 2016) by laying a grid over the image and estimating the presence, relative position, and latent representation of objects in each cell. In GENESIS (Engelcke et al., 2020), the image is recurrently encoded into latent codes per object in a variational framework. In contrast to our method, the above methods do not represent the 3D geometry of the scene explicitly. Recently, Liao et al. (2020) introduced 3D controllable image synthesis to generate novel scenes instead of explaining input views like we do.

**Supervised learning for object instance segmentation, pose and shape estimation.** Loosely related to our approach are supervised deep learning methods that segment object instances (Hou et al., 2019; Prabhudesai et al., 2020), estimate their poses (Xiang et al., 2017) or recover their 3D shape (Gkioxari et al., 2019; Kniaz et al., 2020). In Mesh R-CNN (Gkioxari et al., 2019), objects

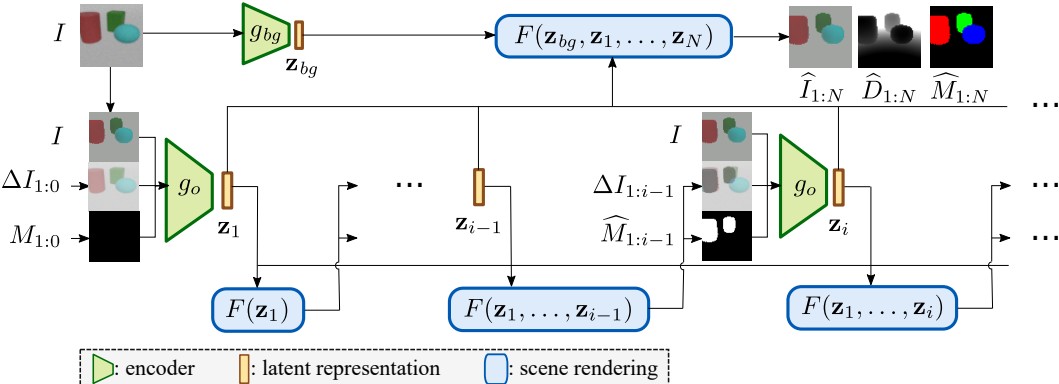

Figure 2: **Multi-object 3D scene representation network.** The image is sequentially encoded into object representations using an encoder network $g_0$. The object encoders additionally receive image and mask compositions $(\Delta I, M)$ generated from the previous object encodings. A differentiable renderer based decoder $F$ composes images and masks from the encodings of previous steps. The background is encoded from the image in parallel and used in the final scene reconstruction.

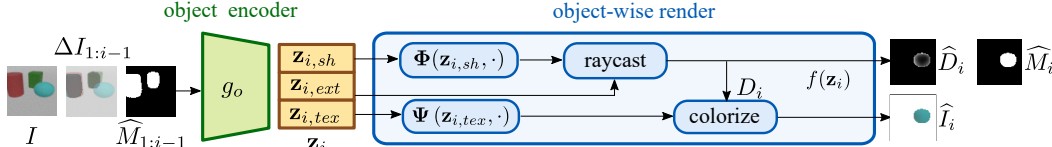

Figure 3: **Object-wise encoding and rendering.** We feed the input image and scene composition images and masks from the previously found objects to an object encoder network $g_o$ which regresses the encoding of the next object $\mathbf{z}_i$. The object encoding decomposes into shape $\mathbf{z}_{i,sh}$, extrinsics $\mathbf{z}_{i,ext}$ and texture latents $\mathbf{z}_{i,tex}$. The shape latent parametrizes an SDF function network $\mathbf{\Phi}$ which we use in combination with the pose and scale of the object encoded in $\mathbf{z}_{i,ext}$ for raycasting the object depth and mask using our differentiable renderer $f$. Finally, the color of the pixels is found with a texture function network $\mathbf{\Psi}$ parametrized by the texture latent.

are detected in bounding boxes and a 3D mesh is predicted for each object. The method is trained supervised on images with annotated object shape ground truth.

**Neural and differentiable rendering.** Eslami et al. (2018) encode images into latent representations which can be aggregated from multiple view points. Scene rendering is deferred to a neural network which needs to be trained to decode the latents into images from examples. Several differentiable rendering approaches have been proposed using voxel occupancy grids (Tulsiani et al., 2017; Gadelha et al., 2017; Jimenez Rezende et al., 2016; Yan et al., 2016; Gwak et al., 2017; Zhu et al., 2018; Wu et al., 2017; Nguyen-Phuoc et al., 2018), meshes (Kato et al., 2018; Loper & Black, 2014; Chen et al., 2019; Delaunoy & Prados, 2011; Ramamoorthi & Hanrahan, 2001; Meka et al., 2018; Athalye et al., 2018; Richardson et al., 2016; Liu et al., 2019; Henderson & Ferrari, 2019), signed distance functions (Sitzmann et al., 2019), or point clouds (Lin et al., 2018; Yifan et al., 2019). Recent literature overviews can be found in (Tewari et al., 2020; Kato et al., 2020). In our approach, we find depth and mask values through equidistant sampling along the ray.

## 3 METHOD

We propose an autoencoder architecture which embeds images into object-wise scene representations (see Fig. 2 for an overview). Each object is explicitly described by its 3D pose and latent embeddings for both its shape and textural appearance. Given the object-wise scene description, a decoder composes the images back from the latent representation through differentiable rendering. We train our autoencoder-like network in a self-supervised way from RGB-D images.

**Scene Encoding.** The network infers a latent $\mathbf{z} = (\mathbf{z}_1, \ldots, \mathbf{z}_N, \mathbf{z}_{bg})$ which decomposes the scene into object latents $\mathbf{z}_i \in \mathbb{R}^d$, $i \in \{1, \ldots, N\}$ and a background component $\mathbf{z}_{bg} \in \mathbb{R}^{d_{bg}}$ where $d, d_{bg}$ are the dimensionality of the object and background encodings and $N$ is the number of objects.

Object are sequentially encoded by a deep neural network $\mathbf{z}_i = g_o(I, \Delta I_{1:i-1}, \widehat{M}_{1:i-1})$ (see Fig. 2). We share the same object encoder network and weights between all objects. To guide the encoder to regress the latent representation of one object after the other, we forward additional information about already reconstructed objects. Specifically, we decode the previous object latents into object composition images, depth images and occlusion masks $(\widehat{I}_{1:i-1}, \widehat{D}_{1:i-1}, \widehat{M}_{1:i-1}) := F(\mathbf{z}_{bg}, \mathbf{z}_1, \ldots, \mathbf{z}_{i-1})$. They are generated by $F$ using differentiable rendering which we detail in the subsequent paragraph. We concatenate the input image $I$ with the difference image $\Delta I_{1:i-1} := I - \widehat{I}_{1:i-1}$ and occlusion masks $\widehat{M}_{1:i-1}$, and input this to the encoder for inferring the representation of object $i$.

The object encoding $\mathbf{z}_i = (\mathbf{z}_{i,sh}^\top, \mathbf{z}_{i,tex}^\top, \mathbf{z}_{i,ext}^\top)^\top$ decomposes into encodings for shape $\mathbf{z}_{i,sh}$, textural appearance $\mathbf{z}_{i,tex}$, and 3D extrinsics $\mathbf{z}_{i,ext}$ (see Fig. 3). The shape encoding $\mathbf{z}_{i,sh} \in \mathbb{R}^{D_{sh}}$ parametrizes the 3D shape represented by a DeepSDF autodecoder (Park et al., 2019). Similarly, the texture is encoded in a latent vector $\mathbf{z}_{i,tex} \in \mathbb{R}^{D_{tex}}$ which is used by the decoder to obtain color values for each pixel that observes the object. Object position $\mathbf{p}_i = (x_i, y_i, z_i)^\top$, orientation $\theta_i$ and scale $s_i$ are regressed with the extrinsics encoding $\mathbf{z}_{i,ext} = (\mathbf{p}_i^\top, z_{\cos,i}, z_{\sin,i}, s_i)^\top$. The object pose $\mathbf{T}_w^o(\mathbf{z}_{i,ext}) = \begin{pmatrix} s_i \mathbf{R}_i^\top & -\mathbf{R}_i^\top \mathbf{p}_i \\ \mathbf{0} & 1 \end{pmatrix}$ is parametrized in a world coordinate frame with known transformation $\mathbf{T}_c^w$ from the camera frame.

We assume the objects are placed upright and model rotations around the vertical axis with angle $\theta_i = \arctan(z_{\sin,i}, z_{\cos,i})$ and corresponding rotation matrix $\mathbf{R}_i$. We use a two parameter representation for the angle as suggested in (Zhou et al., 2019). We scale the object shape by the factor $s_i \in [s_{\min}, s_{\max}]$ which we limit in an appropriate range using a sigmoid squashing function. The background encoder $g_{bg} := \mathbf{z}_{bg} \in \mathbb{R}^{d_{bg}}$ regresses the uniform color of the background plane, i.e. $d_{bg} = 3$. We assume the plane extrinsics and hence its depth image is known in our experiments.

**Scene Decoding.** Given our object-wise scene representation, we use differentiable rendering to generate individual images of objects based on their geometry and appearance and compose them into scene images. An object-wise renderer $(\widehat{I}_i, \widehat{D}_i, \widehat{M}_i) := f(\mathbf{z}_i)$ determines color image $\widehat{I}_i$, depth image $\widehat{D}_i$ and occlusion mask $\widehat{M}_i$ from each object encoding independently (see Fig. 3). The renderer determines the depth at each pixel $\mathbf{u} \in \mathbb{R}^2$ (in normalized image coordinates) through raycasting in the SDF shape representation. Inspired by (Wang et al., 2020), we trace the SDF zero-crossing along the ray by sampling points $\mathbf{x}_j := (d_j \mathbf{u}, d_j)^\top$ in equal intervals $d_j := d_0 + j\Delta d, j \in \{0, \ldots, N-1\}$ with start depth $d_0$. The points are transformed to the object coordinate system by $\mathbf{T}_c^o(\mathbf{z}_{i,ext}) := \mathbf{T}_w^o(\mathbf{z}_{i,ext})\mathbf{T}_c^w$. Subsequently, the signed distance $\phi_j$ to the shape at these transformed points is obtained by evaluating the SDF function network $\mathbf{\Phi}(\mathbf{z}_{i,sh}, \mathbf{T}_c^o(\mathbf{z}_{i,ext})\mathbf{x}_j)$. Note that the SDF network is also parametrized by the inferred shape latent of the object. The algorithm finds the zero-crossing at the first pair of samples with a sign change of the SDF $\mathbf{\Phi}$. The sub-discretization accurate location $\mathbf{x}(\mathbf{u})$ of the surface is found through linear interpolation of the depth regarding the corresponding SDF values of these points. The depth at a pixel $D_i(\mathbf{u})$ is given by the z coordinate of the raycasted point $\mathbf{x}(\mathbf{u})$ on the object surface in camera coordinates. If no zero crossing is found, the depth is set to a large constant. The binary occlusion mask $M_i(\mathbf{u})$ is set to 1 if a zero-crossing is found at the pixel and 0 otherwise. The pixel color $I_i(\mathbf{u})$ is determined using a decoder network $\mathbf{\Psi}$ which receives the texture latent $\mathbf{z}_{i,tex}$ of the object and the raycasted 3D point $\mathbf{x}(\mathbf{u})$ in object coordinates as inputs, i.e. $I_i(\mathbf{u}) = \mathbf{\Psi}(\mathbf{z}_{i,tex}, \mathbf{T}_c^o(\mathbf{z}_{i,ext})\mathbf{x}(\mathbf{u}))$. We speed up the raycasting process by only considering pixels that lie within the projected 3D bounding box of the object shape representation. This bounding box is known since the SDF function network is trained with meshes that are normalized to fit into a unit cube with a constant padding. Note that this rendering procedure can be implemented using differentiable operations which makes it fully differentiable for the shape, color and extrinsics encodings of the object.

The scene images, depth images and occlusion masks $(\widehat{I}_{1:n}, \widehat{D}_{1:n}, \widehat{M}_{1:n}) = F(\mathbf{z}_{bg}, \mathbf{z}_1, \ldots, \mathbf{z}_n)$ are composed from the individual objects $1, \ldots, n$ with $n \leq N$ and the decoded background through z-buffering. We initialize them with the background color, depth image of the empty plane and empty mask. Recall that the background color is regressed by the encoder network. For each pixel $\mathbf{u}$, we

search the occluding object $i$ with the smallest depth at the pixel. If such an object exists, we set the pixel's values in $\widehat{I}_{1:N}, \widehat{D}_{1:N}, \widehat{M}_{1:N}$ to the corresponding values in the object images and masks.

**Training.** We train our network architecture in two stages. In a first stage, we learn the SDF function network from a collection of meshes. The second stage uses the pre-trained SDF models to learn the remaining components for the object-wise scene decomposition and rendering network. We train the SDF networks according to (Park et al., 2019) from a collection of meshes and sample points in a volume around the object and on the object surface. We normalize the size of the input meshes to fit into the unit cube with constant padding $\epsilon = 0.1$.

Our multi-object network architecture is trained self-supervised from RGB-D images containing example scenes composed of multiple objects. To this end, we minimize the loss function

$$L_{total} = \lambda_I L_I + \lambda_D L_D + \lambda_{gr} L_{gr} + \lambda_{sh} L_{sh}, \tag{1}$$

which is a weighted sum of multiple sub-loss functions defined by

$$L_I = \frac{1}{|\Omega|} \sum_{\mathbf{u} \in \Omega} \left\| G\left(\widehat{I}_{1:N}\right)(\mathbf{u}) - G(I_{gt})(\mathbf{u}) \right\|^2 \quad L_D = \frac{1}{|\Omega|} \sum_{\mathbf{u} \in \Omega} \left\| G\left(\widehat{D}_{1:N}\right)(\mathbf{u}) - G(D_{gt})(\mathbf{u}) \right\|$$

$$L_{gr} = \sum_i \max(0, -z_i) + \max(0, -\phi_i(z_i')) \qquad L_{sh} = \sum_i \|\mathbf{z}_{i,sh}\|^2 \tag{2}$$

In particular, $L_I$ is the mean squared error on the image reconstruction with $\Omega$ being the set of image pixels and $I_{gt}$ the ground-truth color image. The depth reconstruction loss $L_D$ penalizes deviations from the ground-truth depth $D_{gt}$. We apply Gaussian smoothing $G(\cdot)$ for which we decrease the standard deviation over time. $L_{sh}$ regularizes the shape encoding to stay within the training regime of the SDF network. Lastly, $L_{gr}$ favors objects to reside above the ground plane with $z_i$ being the coordinate of the object in the world frame, $z_i'$ the corresponding projection onto the ground plane, and $\phi_i(\mathbf{x}_k) := \mathbf{\Phi}(\mathbf{z}_{i,sh}, \mathbf{T}_c^o(\mathbf{z}_{i,ext})\mathbf{x}_k)$. The shape regularization loss is scheduled with time-dependent weighting. This prevents the network from learning to generate unreasonable extrapolated shapes in the initial phases of the training, but lets the network refine them over time.

We use a CNN for both the object and the background encoder. Both consist of a number of convolutional layers with kernel size $(3, 3)$ and strides $(1, 1)$ each followed by ReLU activation and $(2, 2)$ max-pooling. The subsequent fully connected layers yield the encodings for objects and background. Similar to (Park et al., 2019), we use multi-layer fully-connected neural networks for the shape decoder $\mathbf{\Phi}$ and texture decoder $\mathbf{\Psi}$. Further details are provided in the supplementary material.

## 4 EXPERIMENTS

We evaluate our approach on synthetic scenes based on the Clevr dataset (Johnson et al., 2017) and scenes generated with ShapeNet models (Chang et al., 2015). The Clevr-based scenes contain images with a varying number of colored shape primitives (spheres, cylinders, cubes) on a planar single-colored background. We modify the data generation of Clevr in a number of aspects: **(1)** We remove shadows and additional light sources and only use the Lambertian rubber material for the objects' surfaces. **(2)** To further increase shape variety, we apply random scaling along the principal axes of the primitives. **(3)** An object might be completely hidden behind another one. Hence, the network needs to learn to hide single objects. We generate several multi-object datasets. Each dataset contains scenes with a specific number of objects which we choose from two to five. Each dataset consists of 12.5K images with a size of 64×64 pixels. Objects are randomly rotated and placed in a range of $[-1.5, 1.5]^2$ on the ground plane while ensuring that any two objects do not intersect. Additionally to the RGB images, we also generate depth maps for training as well as instance masks for evaluation. The images are split into 9K training, 1K validation, and 2.5K testing examples. For the pre-training of the DeepSDF (Park et al., 2019) network, we generate a small set of nine shapes per category with different scaling along the axes for which we generate ground truth SDF samples. Different to (Park et al., 2019), we sample a higher ratio of points randomly in the unit cube instead of close to the surface. We also evaluate on scenes depicting either cars or armchairs as well as a mixed set consisting of mugs, bottles and cans (tabletop) from the ShapeNet model set. Specifically, we select 25 models per setting which we use both for pre-training the DeepSDF as well as for the generation of the multi-object datasets. We increase the size of the dataset to (18K/2K/5K). The evaluation is performed on two different test sets: (1) with known shapes and (2) with new objects.

Figure 4: **Qualitative results on the Clevr dataset (Johnson et al., 2017) with three and five objects.** Our object-wise scene representation decouples all objects from the background.

**Network Parameters.** For the Clevr / ShapeNet datasets, the object encoding dimension is set to $D_{sh} = 8/16$, and $D_{tex} = 7/15$. The shape decoder is pre-trained for 10K epochs. We decrease the loss weight $\lambda_{sh}$ from $0.025/0.1$ to $0.0025/0.01$ during the first 500K iterations. The remaining weights are fixed to $\lambda_I = 1.0$, $\lambda_{depth} = 0.1/0.05$, $\lambda_{gr} = 0.01$. We add Gaussian noise to the input RGB images. Depth images are clipped at a distance of 12. The renderer evaluates at 12 steps along each ray. Gaussian smoothing is applied with kernel size 16 and decreasing sigma from $\frac{16}{3}$ to $\frac{1}{2}$ in 250K steps. We use the ADAM optimizer (Kingma & Ba, 2014) with learning rate 0.0001 and batch size 8 to train for a dataset-specific number of epochs (see supplementary material for more details).

**Evaluations Metrics.** We evaluate the task of learning object-level 3D scene representations using measures for instance segmentation, image reconstruction, and pose estimation. To evaluate the capability of our model to recognize objects that best explain the input image, we consider established instance segmentation metrics. An object is considered to be correctly segmented if the intersection-over-union (IoU) score between ground truth and predicted mask is higher than some threshold $\tau$. To account for occlusions, only objects that occupy at least 25 pixels are taken into account. We report average precision ($AP_{0.5}$), average recall ($AR_{0.5}$), $F1_{0.5}$-score for a fixed $\tau = 0.5$ as well as the mean AP over thresholds in range $[0.5, 0.95]$ with stepsize 0.05 (Everingham et al., 2010). Furthermore, we list the ratio of scenes were all visible objects were found w.r.t. $\tau = 0.5$ (allObj). Next, we evaluate the quality of both the RGB and depth reconstruction obtained from the generated objects. To assess the image reconstruction, we report *Root Mean Squared Error* (RMSE), *Structural SIMilarity Index* (SSIM) and *Peak Signal-to-Noise Ratio* (PSNR) scores. For the object geometry, we compute similar to (Eigen et al., 2014) the *Absolute Relative Difference* (AbsRD), *Squared Relative Difference* (SqRD), as well as the RMSE for the predicted depth. Furthermore, we report the error on the estimated objects' position (mean) and rotation (median, sym.: up to symmetries) for objects with a valid match w.r.t. $\tau = 0.5$. More details on the metrics are provided in the supplementary material. We show results over five runs per configuration and report the mean.

## 4.1 CLEVR DATASET

In Fig. 4, we show reconstructed images, depth and normal maps on the Clevr (Johnson et al., 2017) scenes. Our model provides a complete reconstruction of the individual objects although they might be partially hidden in the image. The network can infer the color of the objects correctly and gets a basic idea about shading (e.g. that spheres are darker on the lower half) and coarse texture. The shape characteristics such as extent, edges or curved surfaces are well recognized. Our model needs to fill all object slots. We sometimes observed that it fantasizes and hides additional objects behind others. Some reconstruction artifacts at object boundaries are due to rendering hard transitions between objects and background. More results and typical failure cases are shown in the supplementary material. Our 3D scene model naturally facilitates generation and manipulation of scenes by altering the latent representation. In Fig. 1, we show example operations like switching the positions of two objects, changing their shape, or removing an entire object. The explicit knowledge about 3D shape also allows us to reason about object penetrations when generating new scenes. Specifically, we evaluate an object intersection loss $L_{int}$ on the newly sampled scenes to filter out those that turn out to be unrealistic due to an intersection between objects (see supplementary material for details).

**Ablation Study.** We evaluate various components of our model on the Clevr dataset with three objects. In Table 1, we evaluate on training settings where we left out each of the loss functions and also demonstrate the benefit of Gaussian smoothing (denoted by $G$) on the image reconstructions. At the beginning of training, the shape regularization loss is crucial to keep the shape encoder close to the pretrained DeepSDF shape space and to prevent it from diverging due to the inaccurate pose estimates of the objects. Applying and decaying Gaussian blur distributes gradient information in the

Table 1: **Results on Clevr dataset (Johnson et al., 2017).** The combination of our proposed loss with Gaussian blur is essential to guide the learning of scene decomposition and object-wise representations. We highlight best (bold) and second best (underlined) result for each measure. Using different maximum numbers of objects in our network, we further train our model on scenes with 2, 4, or 5 objects. Despite the increased difficulty for larger number of objects, our model recognizes most objects in scenes with two to five objects. Models trained with fewer objects can successfully explain scenes with a larger number of objects (# $obj=o_{train}/o_{test}$).

| | Instance Reconstruction | | | | | Image Reconstruction | | | Depth Reconstruction | | | Pose Est. |
|---|---|---|---|---|---|---|---|---|---|---|---|---|
| | mAP ↑ | AP$_{0.5}$ ↑ | AR$_{0.5}$ ↑ | F1$_{0.5}$ ↑ | allObj ↑ | RMSE ↓ | PSNR ↑ | SSIM ↑ | RMSE ↓ | AbsRD ↓ | SqRD ↓ | Err$_{pos}$ |
| # obj=3/3, w/o $L_I$ | 0.686 | 0.941 | 0.879 | 0.899 | 0.709 | 0.199 | 14.176 | 0.713 | 0.595 | 0.023 | 0.073 | 0.159 |
| # obj=3/3, w/o $L_D$ | 0.023 | 0.086 | 0.076 | 0.078 | 0.008 | 0.085 | 22.142 | 0.837 | 2.745 | 0.231 | 1.061 | 1.341 |
| # obj=3/3, w/o $L_{sh}$ | 0.01 | 0.032 | 0.027 | 0.028 | 0.001 | 0.13 | 17.907 | 0.763 | 1.455 | 0.147 | 0.556 | 0.676 |
| # obj=3/3, w/o $L_{gr}$ | 0.09 | 0.195 | 0.205 | 0.198 | 0.008 | 0.09 | 21.163 | 0.799 | 1.159 | 0.087 | 0.32 | 0.81 |
| # obj=3/3, w/o $G$ | 0.164 | 0.296 | 0.161 | 0.199 | 0.001 | 0.114 | 19.065 | 0.792 | 1.331 | 0.112 | 0.441 | 0.182 |
| # obj=3/3, full | **0.712** | **0.949** | **0.942** | **0.943** | **0.85** | **0.049** | **26.466** | **0.914** | 0.554 | **0.019** | **0.061** | **0.155** |
| # obj=2/2 | 0.782 | 0.977 | 0.963 | 0.967 | 0.928 | 0.039 | 28.389 | 0.941 | 0.432 | 0.012 | 0.04 | 0.138 |
| # obj=4/4 | 0.688 | 0.941 | 0.919 | 0.926 | 0.746 | 0.054 | 25.632 | 0.899 | 0.584 | 0.022 | 0.064 | 0.151 |
| # obj=5/5 | 0.604 | 0.895 | 0.861 | 0.872 | 0.539 | 0.061 | 24.568 | 0.876 | 0.593 | 0.025 | 0.067 | 0.149 |
| # obj=3/2 | 0.756 | 0.974 | 0.969 | 0.97 | 0.942 | 0.041 | 28.011 | 0.937 | 0.452 | 0.013 | 0.044 | 0.14 |
| # obj=3/4 | 0.613 | 0.883 | 0.853 | 0.863 | 0.512 | 0.06 | 24.669 | 0.88 | 0.665 | 0.028 | 0.083 | 0.179 |
| # obj=3/5 | 0.478 | 0.775 | 0.71 | 0.735 | 0.212 | 0.072 | 23.093 | 0.841 | 0.69 | 0.033 | 0.086 | 0.201 |

images beyond the object masks and allows the model to be trained in a coarse-to-fine manner. This helps the model to localize the various objects in the scene. Moreover, the depth loss is essential for learning the scene decomposition. Without this loss, the network can simply describe several objects using a single object with more complex texture. The usage of the ground loss prevents the model from fitting objects into the ground plane. The image reconstruction loss plays only a minor part for the scene decomposition task but is merely responsible for learning the texture of the objects. Using all our proposed loss functions yields best results over all metrics. Remarkably, our model is able to find objects at high recall rates (0.942 AR at 50% IoU).

**Object Count.** We also report results when varying the maximum number of objects in our model in Tab. 1. We train the models with the corresponding number of objects in the dataset. Obviously, it is on average easier for our model to find and describe the objects in less crowded scenes, while it still performs with high accuracy for five objects.

Due to the sequential architecture of our model, it can even be extended for scenes with more objects than that it has been trained for. As we use a shared encoder for all objects, we can simply reset the number of encoding rollouts to the number of objects in the test data. Again, we assume the number of objects to be known. Although our model would be able to hide redundant objects behind already reconstructed ones without this explicit change, it could not reconstruct additional objects. In these experiments, it performs less well than the trained models for the respective object counts. The achieved average recall and allObj measures still indicate that the model is able to detect the objects at good rates. For instance, for # obj=3/5, we find all objects in about 21% cases but overall 71% of the objects according to AR$_{0.5}$. Extended quantitative evaluation as well as qualitative results can be viewed in the supplementary material.

## 4.2 SHAPENET DATASET

Our composed multi-object variant of ShapeNet (Chang et al., 2015) models is more difficult in shape and texture variation than Clevr (Johnson et al., 2017). For some object categories such as cups or armchairs, training can converge to local minima. We report mean and best results over five training runs in Tab. 2, where the best run is chosen according to F1 score on the validation set. Evaluation is performed on two different testsets: scenes containing (1) object instances with shapes and textures used for training and (2) unseen object instances. We show several scene reconstructions in Fig. 5. Further qualitative results are provided in the supplementary material.

For the cars, our model yields consistent performance in all runs with comparable decomposition results to our Clevr experiments. However, we found that cars exhibit a pseudo-180-degree shape symmetry which was difficult for our model to differentiate. Especially for small objects in the

Table 2: **Evaluation on scenes with ShapeNet objects (Chang et al., 2015).** Results for scenes containing objects from different categories. We differentiate between scenes that consist of shapes that were seen during training and novel objects. We report mean and best outcome over five runs.

| | | | Instance Reconstruction | | | | | Image Reconstruction | | | Depth Reconstruction | | | Pose Estimation | |
|---|---|---|---|---|---|---|---|---|---|---|---|---|---|---|---|
| | | | mAP↑ | AP$_{0.5}$↑ | AR$_{0.5}$↑ | F1$_{0.5}$↑ | allObj↑ | RMSE↓ | PSNR↑ | SSIM↑ | RMSE↓ | AbsRD↓ | SqRD↓ | Err$_{pos}$↓ | Err$_{rot}$ [sym.]↓ |
| cars | seen | best | 0.750 | 0.991 | 0.991 | 0.991 | 0.979 | 0.064 | 24.092 | 0.898 | 0.158 | 0.006 | 0.004 | 0.144 | 23.67° [3.29°] |
| | | mean | 0.738 | 0.990 | 0.990 | 0.990 | 0.975 | 0.064 | 23.979 | 0.894 | 0.160 | 0.006 | 0.005 | 0.146 | 22.09° [3.07°] |
| | unseen | best | 0.639 | 0.980 | 0.980 | 0.980 | 0.955 | 0.077 | 22.442 | 0.843 | 0.210 | 0.010 | 0.008 | 0.183 | 24.24° [4.53°] |
| | | mean | 0.632 | 0.977 | 0.977 | 0.977 | 0.944 | 0.077 | 22.454 | 0.842 | 0.208 | 0.010 | 0.008 | 0.184 | 24.25° [4.41°] |
| chairs | seen | best | 0.432 | 0.897 | 0.871 | 0.881 | 0.640 | 0.086 | 21.576 | 0.803 | 0.829 | 0.040 | 0.117 | 0.308 | 43.64° [9.13°] |
| | | mean | 0.329 | 0.642 | 0.638 | 0.640 | 0.188 | 0.102 | 20.137 | 0.772 | 1.021 | 0.058 | 0.196 | 0.296 | 55.12° [7.25°] |
| | unseen | best | 0.377 | 0.852 | 0.821 | 0.833 | 0.534 | 0.092 | 20.994 | 0.778 | 0.890 | 0.052 | 0.137 | 0.395 | 58.79° [10.66°] |
| | | mean | 0.278 | 0.613 | 0.607 | 0.609 | 0.158 | 0.106 | 19.740 | 0.746 | 1.068 | 0.069 | 0.213 | 0.372 | 68.29° [9.28°] |
| tabletop | seen | best | 0.628 | 0.936 | 0.870 | 0.895 | 0.659 | 0.057 | 25.242 | 0.908 | 0.786 | 0.026 | 0.132 | 0.182 | 89.14° |
| | | mean | 0.394 | 0.565 | 0.537 | 0.546 | 0.251 | 0.078 | 22.871 | 0.861 | 1.022 | 0.050 | 0.231 | 0.155 | 88.53° |
| | unseen | best | 0.435 | 0.839 | 0.816 | 0.823 | 0.569 | 0.083 | 21.807 | 0.840 | 1.034 | 0.044 | 0.224 | 0.275 | 89.25° |
| | | mean | 0.285 | 0.530 | 0.521 | 0.523 | 0.237 | 0.102 | 20.160 | 0.800 | 1.172 | 0.061 | 0.291 | 0.238 | 89.99° |

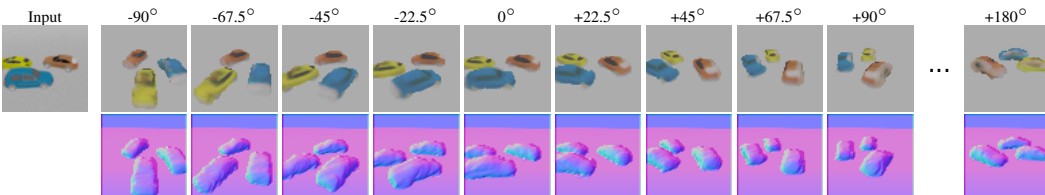

Figure 5: **Qualitative results on ShapeNet (Chang et al., 2015).** Our model obtains a good scene understanding if confronted with more difficult objects (cars, armchairs) and even handles objects from different categories (tabletop scenes with mugs, bottles and cans). It is able to estimate plausible pose and shape of individual objects and learns to decode more complex textures.

background, it favors to adapt the texture over rotating the object. For the armchair shapes, our model finds local minima in pseudo-90-degree symmetries. The median rotation error indicates better than chance prediction for the correct orientation. Rotation error histograms can be found in the supplementary material. For approximately correct rotation predictions, we found that our model was able to differentiate between basic shape types but often neglected finer details like thin armrests which are difficult to differentiate in the images. Our tabletop dataset provides another type of challenge: the network needs to distinguish different object categories with larger shape and scale variation. For this setting, we added further auxiliary losses to penalize object positions outside of the image view as well as object intersections (see supplementary material for details). Our model is able to predict the different shape types with coarse textures. On scenes with instances that were not seen during training, our model often approximates the shapes with similar training instances.

Due to the learned 3D structure, our model is able to render novel views from a scene given a single image (see Fig. 6). Although our model never saw multiple views of the same scene during training and is not tuned for this task, we obtain reasonable results for both scene geometry and appearance. We observe a lower reconstruction accuracy for invisible scene parts, especially for the texture.

Figure 6: **Novel view renderings.** Our model is able to generate new scene renderings for largely rotated camera views from just a single input RGB image. While we noticed a reduced texture accuracy for unseen object parts compared to visible parts, the normal maps are generally good and demonstrate that our model obtains a good 3D structural understanding of the scene.

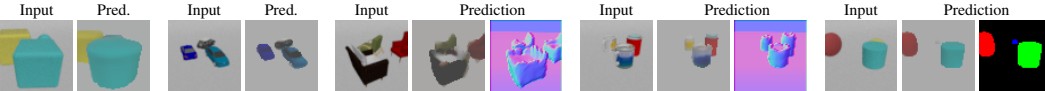

Figure 7: **Evaluation on real images.** We show preliminary results on real images by our model that was trained on synthetic data. We notice that our model is able to capture the coarse scene layout and shape properties of the objects. However, challenges arise due to domain, lighting, camera intrinsics and view point changes indicating interesting directions for future research.

Figure 8: **Limitations**. Input and output pairs for typical failure cases and limitations of our method due to ambiguities for self-supervised learning. See text for details.

We further evaluated our model on real images of toy cars and building blocks (see Fig. 7) for which we adjusted brightness and contrast to visually match the background color of the synthetic data. Note while the scene perspective, camera and image properties are different, our model is able to decompose the scene in these examples into the individual objects and obtain a coarse understanding about their shape and appearance without any further fine-tuning on the new data domain.

**Limitations.** We show typical failure cases of our approach in Fig. 8. Self-supervised learning without regularizing assumptions leads typically to ill-conditioned problems. We use a pre-trained 3D shape space to confine the possible shapes, impose a multi-object decomposition of the scene, and use a differentiable renderer of the latent representation. In our self-supervised approach, ambiguities can arise due to the decoupling of shape and texture. For instance, the network can choose to occlude the background partially with the shape but fix the image reconstruction by predicting background color in these areas. Rotations can only be learned up to a pseudo-symmetry by self-supervision when object shapes are rotationally similar and the subtle differences in shape or texture are difficult to differentiate in the image. In such cases, the network can favor to adapt texture over rotating the shape. Depending on the complexity of the scenes and the complex combination of loss terms, training can run into local minima in which objects are moved outside the image or fit the ground plane. Currently, the network is trained for a maximum number of objects. If all objects in the scene are explained, it hides further objects which could be alleviated by learning a stop criterion.

## 5 CONCLUSION

We propose a novel deep learning approach for multi-object scene representation learning and parsing. Our approach infers the 3D structure of a scene in RGB images by recursively parsing the image for shape, texture and poses of the objects. A differentiable renderer allows images to be generated from the latent scene representation and the network to be trained semi-supervised from RGB-D images. We represent object shapes by signed distance functions. To confine the search space of possible shapes, we employ pre-trained shape spaces in our network. The shape space is represented by a deep neural network using a continuous function representation. Our experiments demonstrate that our model achieves scene parsing for a variety of object counts and shapes. We provide an ablation study to motivate design choices and discuss assumptions and limitations of our approach. We further demonstrate the advantages of our model to reason about the underlying 3D space of a seen scene by performing explicit manipulation on the individual objects or rendering novel views. To the best of our knowledge, our approach is the first to jointly learn the tasks of object instance detection, instance segmentation, object pose estimation, and inference of 3D shape and texture in a single RGB image in a semi-supervised way. We believe our approach provides an important step towards self-supervised learning of object-level 3D scene parsing and generative modeling of complex scenes from real images. Our work is currently limited to simple scenes with few objects on a uniformly colored background. The usage of such synthetic data allows us to evaluate the individual design choices of our model in a controlled setup. In future work, we plan to address challenges of more complex scenes with more diverse background and objects and real imagery.

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
