# OpenReview forum: "Semi-Supervised Learning of Multi-Object 3D Scene Representations"
_ICLR.cc/2021/Conference — Reject_

### Official Review · AnonReviewer1 · 2020-10-29
**Review of paper #794**

**Rating:** 6
**Confidence:** 3

**Review:**

### Weaknesses

- All experiments are synthetic. Although synthetic experiments are common in the field, I see this as a weakness because the method heavily relies on image reconstruction loss on rendered objects. Real scene are very different due to both high level (more clutter, occlusion, number of objects) and low level (shading, less predicable texture) differences from the examples used in the synthetic experiments and will likely require different constraints to solve. The model has access to dense ground truth depth maps too, which is difficult to obtain. The motivation for self-supervised learning is to take advantage of unlabeled in-the-wild images, so considering all of those things, I think the synthetic RGB-D assumption hurts this paper more than an average paper.

- There's much to learn from the object count experiment (Table 1 and Page 6), but I found the results confusing. According to Page 5 which says "each dataset contains scenes with a specific number of objects which we choose from two to five", the network was trained to model a specific number of objects separately in each experiment. If you evaluate a three-object trained network on images containing two objects ("obj=3/2" in Table 1, 10th row), one would expect it to suffer from false positives, but the quantitative results do not show this. The results are almost the same as "obj=3/3". Why is this? Did it learn to solve this problem, or were they somehow not included in the evaluation? I think there's some resemblance to EM, and it would matter a lot if you knew the number of true clusters beforehand, and how over or under-estimations are handled, in understanding the model's behavior.

- Lack of basic baselines. Currently there's no way to know how difficult this dataset is, and how well a fully supervised model would perform.

-  Potentially missing details, additional questions, and suggestions:
     - In RGB reconstruction loss and evaluation, do you also count the background, or is this on ground truth object region only?
     - Page 2 says "By representing 3D geometry explicitly, our approach naturally respects occlusions and collisions between objects and facilitates manipulation of the scene within the latent space." This part was confusing to me. How does it avoid collision of objects in 3D? My understanding is that this is not part of equation 1.
     - Figure 2 is confusing. I think a few captions in the image would have helped a lot.
     - In Table 2, would you attribute the gap between seen and unseen instances more to the shape, or the texture of the object?

- One suggestion is, would you be able to render the scene from a novel viewpoint, since you know the depth, shape and pose of each object? And then evaluate against ground truth rendering from that viewpoint? Being able to do such an experiment could make the synthetic setting more useful.

### Strengths

- Ablation study in Table 1 shows that all of the loss terms are important. This justifies the design and I would consider it an insightful discovery.  For example, the ground truth depth already globally provides sharp object boundaries, but in order to reproduce them, both shape and ground plane loss terms were needed. So the model provides the right constraints in which all of those factors are considered.

- "How do you turn single-object 3D reconstruction models into priors, for scene-level inference?" is still a very open question, and more research is needed in this area. The fact that this methods requires such a prior could be a weakness to someone who prefers to see less assumptions made in favor of generalizability. But I believe objects have good enough scientific foundation to be beneficial in the long run.

### Justification of rating

Despite the weaknesses I listed, I feel that this paper is above threshold, albeit marginal. Other researchers can still learn from the design choices made even if it turns out that those constraints don't work well in real scenes. I understand that this is a difficult task, and having a synthetic model as a reference point would ultimately help, assuming the code for reproducing the experiments will be fully released, to serve as a baseline.

---

> ### Author Response · Authors · 2020-11-18
> **Response to Reviewer 1**
>
> We thank you for your detailed feedback and helpful suggestions. We appreciate that you find our work to be an insightful baseline for the difficult task of multi-object 3D scene representation learning albeit its current limitation to synthetic data.
>
> In this response, we would like to answer your questions and address your concerns.
>
> - Synthetic Data
> Our synthetic scenes are composed of several simple and complex-shaped textured objects on a simple planar background. With this structured data we develop and demonstrate a basic novel learning architecture and point out the effectiveness of our design choices. The available ground truth information of our synthetic data allows us to perform an extensive evaluation in a defined setting which yields valuable insights.
> We expect significant advances in the upcoming years on the modeling of more realistic scenes by differentiable renderers. Our method tackles the orthogonal task of decomposing scenes into an object-wise 3D description in a self-supervised way. In this regard, our approach provides novel insights. We are optimistic that our work can provide a basis for object-wise representation learning of more complex and more realistic scenes using potential future advances in differentiable rendering.
> Regarding unlabeled in-the-wild images, we argue that the widely available RGB-D cameras could provide the necessary depth images for the training losses. The trained model could be used on RGB camera images since it does not require depth images at test time. Hence, the requirement of RGB-D data during training does not harm the motivation for our approach.
>
> - Object Count Experiment
> In the object count experiments, we set the number of object slots in the network according to the known number of objects in both the train and test data. As we use a shared encoder for all objects, we set the number of encoding rollouts to the current number of objects in the scene. While our network is able to hide remaining objects behind already reconstructed objects, it would not be able to reconstruct additional ones if we would not reset the number of encoding steps. We clarified this in the updated version.
>
> - Supervised Baseline
> We want to point out that using supervision in our approach is not straightforward due to the ambiguity of the objects' order. To apply supervision e.g. for object poses, there would be two options: We would either need to tell the network which object to choose at the current step (e.g. using object masks as additional input channel) or we would need to match predicted and ground-truth objects. The first option would need additional information on the object id also during inference and thus neglects to learn to decompose a scene on its own. For the second option we evaluated using the distance between objects' 3d positions for measuring the matching, however, this did not lead to stable training.
>
>
>
> - Additional Comments
> 	- We do include the background in our RGB evaluation since the reconstructed object might be rendered on a wrong position and thus occlude GT background area which should also be penalized.
> 	- The statement refers to the intersection loss L_int (details in supplementary) which we use for the following settings:
> 	  a) as extension to the training objective when training on tabletop scenes
> 	  b) for validation of manipulated scenes where we sample new object positions
>         We clarified this sentence.
> 	- We updated Figure 2 in our main paper to make its structure more clear and easier to understand.
> 	- Table 2:
>     In our experiments, we find that it is depending on the object category whether the gap between seen and unseen objects would depend more on shape or on texture.
>     Specifically, we considered the relative change in the RGB and depth losses which would indicate higher changes compared to GT in texture and shape, respectively.
>     For cars, we observed a stronger relative worsening of the depth loss which indicates that the model prefers to focus on the new object's texture.
>     However, it is the other way round for the tabletop scenes.
>     There are two possible explanations for this: One the one hand, higher variance in texture for tabletop objects makes it more difficult to replace new objects with similar known ones. On the other hand, the higher variance in shapes results in a lower relative increase of the depth loss for new shapes.
>     We did not find such clear trends for the chair scenes.
>
>
> - Novel View Rendering
> We added qualitative results (Figure 6) with renderings from novel views. While we want to highlight that our model was never trained for this task specifically, we get reasonable results.
>
> We will release our code, trained models, and datasets to foster reproducibility and future research.

---

### Official Review · AnonReviewer2 · 2020-10-31
**Interesting work on 3D scene inference but has space for improvement**

**Rating:** 6
**Confidence:** 4

**Review:**

The authors proposed a method for 3D scene inference, which jointly does object instance detection, instance segmentation, object localization, and 3D shape and texture inference. The authors designed an autoencoder-like network such that it can be trained in a self-supervised way from RGBD images. Some limitations:
- The authors claimed they are the first to do these tasks jointly, and they did not compare their method to any existing work, but only tested on the Clever and ShapeNet dataset. However, there are existing depth/normal estimation algorithms which can serve as baselines.
- It is not justified the superiority of the design of the method. Did it model the relationships between objects? Why not using an object detector combined with a 3D shape/texture reconstruction neural network?
- The authors did not test their method on real data. Only tested on the Clever and ShapeNet dataset.
- The pipeline figure (Figure 2) is very difficult to understand. Might be better with more text illustrations in the figure, not only notations.

---Post-Rebuttal---

Thank the authors for their response. I now agree with the authors and other reviewers that the authors' approach has its novelty (self-supervised, rendering, etc.), and the ablation study in Table 1 is reliable to prove each component is useful. Therefore, I increase my rating by 1.

---

> ### Author Response · Authors · 2020-11-18
> **Response to Reviewer 2**
>
> We thank you for your time in reviewing our work and pointing out potential issues. We want to address these points in the following:
>
> - Missing comparison to existing work, especially depth/ normal estimation methods:
> Our method does not only generate a depth-map based 3D reconstruction of a scene but also obtains an understanding of the scenes on object-level. This means, our model yields a disentangled representation for each object that can be used both to reason about them as well as to generate new scenes where individual objects were explicitly manipulated. This aspect of our work is motivated by potential future fields of application like robotics where dealing with objects is a crucial part of many tasks.
> From our perspective, a comparison to methods that focus only on a single part of the presented task would therefore ignore the additional value of our model and thus would yield only limited additional insights.
> We are happy for suggestions of particular meaningful baselines which we could try to compare with in the final version.
>
> - Design of method
> We have carefully designed our network architecture and training losses to tackle the task of self-supervised learning of multi-object 3D scene representation. Our main design choices are to use a pretrained 3D shape embedding and a differentiable renderer which encodes prior knowledge of object variation in a scene and how scene observations are generated from the latent representation. Differentiable rendering facilitates self-supervised training of the scene decomposition network.
> To the best of our knowledge, our approach is the first to tackle this challenging problem.
> Object bounding box detection pipelines such as Yolo or Faster R-CNN are typically trained supervised using ground-truth annotation of objects. Our encoder recursively detects and describes objects in the scene in terms of pose, shape and texture and is trained self-supervised.
>
> - Object relations are modeled in the following ways in our approach:
>   1. The used ground loss L_gr motivates the network to place objects above the ground plane. For this it considers both their 3D position as well as their shape. As shown in the ablation study, this loss is important to learn a suitable scene representation.
>   2. In the tabletop scenes, we include relations between objects by an intersection loss (see the supplementary material). Specifically, we penalize intersection between the objects' 3D shapes. This is made possible by explicitly reasoning about the underlying 3D structure of the scene.
>   3. We can reason on object intersections when modifying object latents and generating new scenes, which we also used for the qualitative results shown for scene manipulation.
>
> - Evaluation on real data
> We provided qualitative results on real Clevr-like scenes in the supplementary material of the submitted version. In the updated version, we further included reconstructions of real scenes that consist of toy cars (Figure 7, main paper).
> For these experiments, we used the models trained and evaluated on synthetic data in the paper. Our model is able to obtain a basic understanding of the scene's structure, hence, it provides a basis for future research on scene parsing in real imagery.
>
> - Pipeline Figure
> We updated Figure 2 in our main paper to make its structure more clear and easier to understand.

---

### Official Review · AnonReviewer4 · 2020-11-03
**Interesting idea for scene parsing; more discussions on real-world use needed**

**Rating:** 6
**Confidence:** 4

**Review:**

Summary:
This paper introduces a new model for obtaining object-level 3D scene representations from images. The proposed method models 3D scenes by parsing objects one-by-one into structured representations of shape, texture, and poses. A differentiable renderer incorporated in this pipeline enables self-supervised training as well as test-time image generation.

The method is interesting and novel, though general ideas are borrowed from Park et al. 2019, Sitzmann et al. 2019 and Wang et al. 2020. The paper is well-written and easy to follow for the most part. Also, the experiments performed have supported the claims from the authors.

Pros:
- This is the first work to jointly learn the tasks of object instance detection, instance segmentation, object localization, and inference of 3D shape and texture in a single RGB image through self-supervised scene decomposition.
- The idea of incorporating a renderer into the parsing pipeline is interesting and meaningful. In this way, the method is able to perform self-supervised decomposing of the scene.
- The authors perform extensive experiments and ablation studies on the clevr and ShapeNet dataset, which have proved the effectiveness of the proposed pipeline on these datasets and provided useful information on designing such kind of pipeline.

Cons:
- Although the authors provide preliminary results on real-world images, the extension of this pipeline to real-world scenes is doubtful:
  - The number of objects is fixed/given for the framework; the increasing number of objects in complex scenes leads to more ill-conditioned problems, which will make the self-supervised decomposition extremely hard.
  - Scene conditions (e.g. light) are not modeled or overly simplified (e.g. background) within this pipeline, and the differentiable renderer used (or even the sota differentiable renderers) are not able to sufficiently model complex real scene conditions.
  - It would be appreciated if you could include more complex shapes in real-world images. For example chairs/toy cars as in the ShapeNet.

Minor Comments & Questions:
- The `Change Shape` example for ShapeNet in Figure 1 does not seem to be obvious.
- Figure 2 is not clear enough at first glance. You can add some texts in Figure 2 to make it more self-contained and understandable, e.g. "g_O: object encoder", "g_{bg}: background encoder", etc. Also, the `Top` and `Bottom` is clearly separated. You should consider to split and reorganize these two figures.
- The method section is slightly verbose and can be better organized.
- Is it possible to render new views given the parsed scene representation? This seems to be possible but not mentioned in the paper.

---

> ### Author Response · Authors · 2020-11-18
> **Response to Reviewer 4**
>
> Thank you for your insightful feedback and constructive comments.
> We are happy that you appreciate our idea of using a differentiable renderer to learn object-level 3D scene representations in a self-supervised way. We believe that our work will provide valuable insights for future work in this direction.
> In the following, we would like to answer your questions and address your concerns:
>
> - Fixed/given number of object:
> The main focus of this work is to demonstrate the feasibility of learning multi object scene representations with latent 3D representations of the objects.
> We note that our method could for instance be extended to varying number of objects by predicting a stop variable for recursive parsing in future work. This is also supported by our object count experiment. We demonstrated that one advantage of our model is its easy adaptability to a different number of objects than it was trained for. Please also note the improved explanation in the corresponding section.
> Hierarchical scene parsing could be another interesting avenue of future research to handle more complex scenes. Nevertheless, our work provides a base study in these directions and can serve as a reference.
>
> - Simple scene conditions:
> As also pointed out by the reviewer, research on differentiable rendering is still in an early stage. We expect significant advances in the upcoming years on the modeling of more realistic scenes for differentiable renderers. Our method tackles the orthogonal task of decomposing scenes into an object-wise 3D description in a self-supervised way. In this regard, our approach provides novel insights. We are optimistic that our work can provide a basis for object-wise representation learning of more complex and more realistic scenes using potential future advances in differentiable rendering.
>
> - Additional evaluation on real world scenes:
> We expand the qualitative evaluation of real world scenes with toy car scenes. Additional results can be found in Figure 6 of the main paper (updated version).
>
> - Further Comments
> 	- We marked the changed object of the 'change shape' example in Figure 1 to make the differences clearer.
> 	- We updated Figure 2 in our main paper to make its structure clearer and easier to understand.
> 	- We extended the description of the method.
> 	- We added a new Figure with renderings from novel views from a single image to the main paper.

---

### Author Response · Authors · 2020-11-18
**Updates in Paper**

We thank all reviewers for their time and insightful constructive comments.

We updated our main paper and the supplementary material to address the concerns and points that were raised.
Most importantly, we did the following changes:
  - We present qualitative results for novel view renderings in both the main paper (Fig. 6) and the supplementary material (Fig. 10).
  - We added additional qualitative results on real images in the main paper including either building blocks or toy cars (Fig. 7).
  - We revised the description of the method and clarified our proceeding in the object count experiment.
  - We updated Figure 2 (network pipeline) according to suggestions of the reviewers.

Please further refer to our individual responses to the reviewers.

---

### Decision · Program_Chairs · 2021-01-07
**Final Decision**

**Decision:**

Reject

**Comment:**

The paper proposes learning of 3D object representation from images. The pretraining used assumes it can generate implicit 3D models for the objects, and then objects are detected in multi-object scenes without further supervision. Reviewers raised concerns regarding experiments being conducted only on synthetic data. Authors are encouraged to try out their approach on real data, to demonstrate the benefits of their solution.